# Microwave-Assisted Conversion of Carbohydrates

**DOI:** 10.3390/molecules27051472

**Published:** 2022-02-22

**Authors:** Leonid M. Kustov, Alexander L. Kustov, Tapio Salmi

**Affiliations:** 1Chemistry Department, Moscow State University, 1 Leninskie Gory, Bldg. 3, 119991 Moscow, Russia; kyst@list.ru; 2N.D. Zelinsky Institute of Organic Chemistry RAS, 47 Leninsky Prosp., 119991 Moscow, Russia; 3Faculty of Science and Engineering, Abo Akademi University, 3 Tuomiokirkontori, FI-20500 Turku, Finland; tapio.salmi@abo.fi

**Keywords:** catalysis, carbohydrate wastes, microwave irradiation, 5-hydroxymethylfurfural, alcohols

## Abstract

Catalytic conversion of carbohydrates into value-added products and platform chemicals became a trend in recent years. Microwave activation used in the processes of carbohydrate conversion coupled with the proper choice of catalysts makes it possible to enhance dramatically the efficiency and sometimes the selectivity of catalysts. This mini-review presents a brief literature survey related to state-of-the-art methods developed recently by the world research community to solve the problem of rational conversion of carbohydrates, mostly produced from natural resources and wastes (forestry and agriculture wastes) including production of hydrogen, synthesis gas, furanics, and alcohols. The focus is made on microwave technologies used for processing carbohydrates. Of particular interest is the use of heterogeneous catalysts and hybrid materials in processing carbohydrates.

## 1. Introduction

Conversion of carbohydrates produced from biomass into value-added products and platform chemicals recently became a global challenge [1,2,3,4,5,6,7]. The environmental aspect of the problem is also important. Municipal solid wastes containing a large pool of polysaccharides (basically cellulose) should be utilized and recycled.

The products that can be produced selectively from carbohydrates may be gaseous, liquid, and solid. The conversion into gas products (synthesis gas, hydrogen) may be very interesting because of the increasing interest to hydrogen as an energy carrier. This short review is focused on novel energy-efficient and green approaches to conversion of carbohydrates, with catalysis being used as the most efficient technology for the selective production of target chemicals. The biochemical (enzymatic) conversion of carbohydrates will not be considered here. The focus will be made on the use of microwave treatment in carbohydrates valorisation. The existing technologies mostly reached the saturation level in their development and the further progress becomes possible by using non-traditional approaches, such as the non-equilibrium conditions, gradient technologies, combination of endothermic and exothermic processes, ex-situ or in-situ application of electromagnetic activation, such as microwave activation, low-temperature plasma, etc. The majority of results available in the literature on the microwave effects in the catalytic processes of carbohydrates conversion provides a convincing evidence that microwave activation can enhance the reaction rate and change the product selectivities, while decreasing the reaction temperature [1,2,3,4,5,6,7]. However, the barriers to overcome on the way to industrial application of microwave irradiation in such catalytic processes are related to the problem of scaling up. Also, there are quite many examples on the positive effects of the microwave plasma, but the microwave plasma (unlike corona discharge and some other types of plasma) requires the use of reduced pressures (typically 10–50 Torr), which suppresses significantly the space time yield of the desired products and thereby limits the possibilities for upscaling.

Also, in mastering the technology for microwave -assisted carbohydrates processing, one should consider that components of biowastes and most products (liquid or gaseous) are not absorbing the microwave irradiation, so the only option is to use the catalysts that are capable of strongly interacting with the microwave radiation. The Maxwell–Wagner interphase polarization mechanism can efficiently operate in mixed oxide catalysts and supported catalysts. If microwave activation is used in catalytic processes, it should be considered that the objects of study are nonuniform multiphase systems including catalysts and liquids/gases (that can dissociate or can be polarized at the catalyst surface) in which volumetric structural and property changes occur under the action of electromagnetic radiation. The main idea of microwave catalysis is to exert volumetric controllable electromagnetic action on the catalyst–reagent system. This action should change the state of the system and increase the effectiveness of the work of catalysts, the selectivity of substrate conversion into valuable products, and catalyst stability. One of the main directions toward solving this problem is a decrease in the temperature by process execution under mild conditions in an electromagnetic microwave field.

Finally, considering an avalanche-like growth of the literature on carbohydrates processing in recent decade, the mini-review will mainly include results and approaches published in the last 5–10 years, while earlier publications will be considered only if they provide valuable information.

## 2. Catalytic Processes for Conversion of Carbohydrates Based on the Use of Microwave Irradiation

Diverse sources of radiation (microwave or ultrasound, irradiation by UV or IR light) can be used in catalytic conversion of carbohydrates. The effects resulted from microwave irradiation with the frequency ranging from 200 to 300 GHz are of particular interest, although generally the frequency of 2.45 GHz is used [1,2,3,4,5,6,7]. The benefit of microwaves is the rapid and selective energy supplied to molecules/species that absorb microwaves. This energy is also transformed into heat. As a result, fast supply of energy into the reaction volume can be realized. Microwave technology demonstrated advantages over the thermal pyrolysis [8]. This technology increases the rate of reactions by rapid energy-efficient heating, increased productivity (or space–time yield) and lower production costs [9,10]. The most interesting effects of the occurrence of temperature gradients and non-equilibrium conditions are observed when the reaction medium or material (a catalyst) consists of several phases with different microwave radiation absorption ability (dielectric tangent loss). The microwave treatment allows a decrease of the reaction time, an increase of the yields and selectivities, and a decrease of the energy and reagent consumption [9,10].

A catalyst capable of absorbing microwave energy or a microwave-absorbing material (SiC, carbon materials) introduced into the reaction volume can be heated. The dielectric polarization contributes significantly to the mechanism of the microwave activation due to the presence of water and other polar compounds in the feed or polar functional groups in the molecules of carbohydrates. The second mechanism involves free charges excited in solids and contributes to heating because of the Ohmic loss; this mechanism is characteristic of conductive or semiconductive catalysts. So, in the case of carbohydrate conversion, all of the mechanisms can be efficient if the system contains polar molecules and solid catalysts containing supported metals or transition metal oxides or carbon materials.

Numerous literature data show that, under microwave activation conditions, higher reaction rates are achievable than under thermal activation [11,12,13,14,15,16,17,18,19,20,21,22,23,24,25,26,27,28,29,30,31,32,33,34,35,36,37,38,39,40,41,42]. This is the way to enhance the yields of final products. The most illustrative examples of the use of microwave activation in the conversion of carbohydrates will be discussed here.

Currently, the choice of an appropriate catalyst for a heterogeneous catalytic process of carbohydrate conversion that is ideally suitable for activation in microwave fields is not quite a simple task. The catalyst should exhibit significant catalytic activity and, at the same time, possess optimal electrophysical properties (dielectric loss tangent and conductivity). To obtain a maximum positive effect of microwave irradiation on a catalytic system, comparatively inert components strongly interacting with electromagnetic fields may be introduced into the catalysts. For instance, zeolites that are known to catalyze the conversion of carbohydrates are not microwave-absorbing materials, thus, other catalytic systems should be designed to properly understand the benefits of the microwave effects.

The microwave activation can be used more efficiently if the active catalyst components are placed on a support that is not a microwave absorbing material. This will create superheated regions (hot spots) only in the volume or on the surface of the active catalyst phase. This provides the possibility of eliminating side reactions and suppressing energy expenses for heating inert materials that do not participate in the catalytic process.

Bifunctional catalysts, in which different catalytic components are responsible for different stages of the process, should be considered separately. A typical example is transformation of an organic molecule on metal nanoparticles supported on acid or basic supports. A metallic component activates the initial molecule via breaking bonds and/or charge transfer (dehydrogenation, hydrogenolysis), whereas other transformations (alkylation, isomerization) occur on acid/basic centers.

### 2.1. Synthesis of 5-Hydroxymethylfurfural

5-Hydroxymethylfurfural (5-HMF) derived from biomass is considered nowadays as one of the so-called platform chemicals (Figure 1).

The data on the synthesis of 5-HMF under microwave activation are presented in Table 1. It is seen that the use of polar catalysts and transition metal containing catalysts (Ti, Sn, W, etc.) provides a gain in the performance compared to the thermal mode of operation as well as a significant decrease of the reaction time to a few minutes. The microwave-stimulated fast conversion of carbohydrates into 5-HMF catalyzed by mesoporous TiO_2_ nanoparticles was reported by Dutta et al. [11]. Monosaccharides (D-fructose, D-glucose) and disaccharides (sucrose, maltose, cellobiose) were converted into 5-HMF in aqueous or organic media. The influence of the solvent nature, microwave parameters, catalyst mass, reaction time, and substrate nature on the yields of 5-HMF was investigated. The high surface area, the presence of coordinatively unsaturated cations (Lewis acid sites) and uniform structure of titania nanoparticles were beneficial for the high activity in dehydration of carbohydrates. For comparison, the commercial sample of TiO_2_ is not active in this process. The higher value of the dielectric tangent loss (tan δ) of the solvent (DMSO) resulted a higher yield of 5-HMF in organic media than in water. The catalyst can be recycled four times without any loss of the activity. The same authors proposed a hierarchical macro/mesoporous titanium phosphate prepared by a slow evaporation method by using titanium isopropoxide and orthophosphoric acid as sources of titanium and phosphorus and pluronic P123 as the structure directing agent [12]. This material also shows excellent catalytic activity in the microwave-assisted conversion of carbohydrates into 5-HMF. These authors also tested large-pore mesoporous tin phosphate prepared using Pluronic P123 in the synthesis of 5-HMF from carbohydrates (fructose, glucose, sucrose, cellobiose, and cellulose) in H_2_O-MIBK (methyl-isobutylketone) biphasic solvent [13]. The yields of 5-HMF of 77, 50, 51, 39, and 32%, respectively, were obtained from fructose, glucose, sucrose, cellobiose, and cellulose under microwave heating at 423 K. This result shows the benefits of large mesopores in catalytic reactions that involve bulky carbohydrate molecules.

Microwave-assisted synthesis of 5-HMF from glucose with AlCl_3_ as a catalyst is reported [14]. Glucose isomerization and dehydration occur under microwave treatment with a high 5-HMF yield (70%) in a biphasic solvent mixture (water/THF). The reaction takes a few minutes at 165 °C.

Hybrid nanomaterials are most interesting for the application in carbohydrates processing, since such materials combine advantages and demonstrate a synergy of components of such a hybrid nanomaterial. The advantages over the non-hybrid materials may be related to a combination of meso- and macroporosity, as well as to a combination of the presence of several functions (like acid and base functions, metal oxide or metal nanoparticle functions) exhibiting different but in some case complementary catalytic properties. The synergy of hybrid nanomaterials, inter alia, may originate from the strengthening of the functional groups of one component of the hybrid material due to the presence of another component. Bifunctional crystalline microporous organic polymers containing sulfonic acid and secondary amine groups representing an example of hybrid materials were synthesized by the reaction of 2,2′-benzidinedisulfonic acid and 4,4′-diaminostilbene-2,2′-disulfonic acid with cyanuric chloride [15]. They were found to be efficient heterogeneous catalysts for the synthesis of 5-HMF from carbohydrates under microwave irradiation.

Another type of bifunctional hybrid materials used as catalysts, a highly ordered 2D-hexagonal mesoporous SBA-15 organosilica with the surface area of 652 m^2^ g^−1^ and average pore diameter of 9.4 nm, was demonstrated to be active in the synthesis of 5-HMF under microwave conditions [16]. The highest yield of 74% from fructose was observed in a polar solvent dimethyl sulfoxide under microwave heating.

Microwave-assisted conversion of derivatives of fructose, glucose and cellulose to 5-HMF with tungstophosphoric acid encapsulated dendritic fibrous mesoporous silica as a hybrid catalyst was demonstrated by Vasudevan et al. [17]. Fructose dehydration proceeds with the yield of 92% 5-HMF and full conversion of fructose at 120 °C for 30 min under the microwave heating in THF. In the case of glucose and cellulose, the 5-HMF yield was 58% and 16.2%, respectively, in DMSO.

Sulfuric acid and montmorillonite clay were also used as catalysts for 5-HMF and levulinic acid (Figure 2) synthesis from carbohydrates under microwave radiation [18]. The reaction time was reduced 3–5 times by using the microwave mode of operation.

Diverse carbohydrates (fructose, glucose, sucrose, and cellulose) were studied in the conversion into 5-HMF using ZrMo mixed oxides modified by carboxylic acids (stearic acid, palmitic acid, myristic acid, and lauric acid) under thermal and microwave conditions [19]. In spite of the better performance of the catalyst under microwave mode of operation, in particular, reduced time of the reaction and lower reaction temperatures, the preference was given to the thermal mode because of the easier upscaling and potential commercialization of the process.

Catalytic isomerization of D-glucose to D-fructose over hierarchical BEA zeotypes in water under microwave activation at 75 °C was also studied [20].

### 2.2. Synthesis of Furfural and Derivatives

Furfural can be easily produced from xylose, water-insoluble hemicelluloses, and a water-soluble fraction of corncob via a tin-loaded montmorillonite solid acid catalyst (Table 1) [21]. The presence of Lewis acids and Brønsted acids in the catalyst improves the furfural yield (76.79%) and selectivity (82.45%) at 180 °C for 30 min. The use of microwave heating further enhanced the performance of the catalyst.

Hydrolysis of xylan, as well as the conversion of D-xylose and D-lyxose and furfural production with molybdate as a catalyst were found to be more efficient under microwave irradiation [22]. The reaction time under microwave irradiation was reduced 400 times in comparison with the thermal process. The microwave-stimulated hydrolysis of polysaccharides was also performed onto polyoxymetalates as catalysts [23]. Polyoxymetalates are good microwave-absorbing materials, which are favorable in the hydrolysis of cellulose. Phosphotungstic and silicotungstic acids demonstrated high activity and appropriate recyclability, unlike phosphomolybdic acid. The energy consumption in hydrolysis can be reduced by up to 23% using microwave activation.

Fructose conversion into 5-HMF in ionic liquids was studied under microwave activation without any catalysts [24]. The yield of 5-HMF up to 97% was obtained after 3 min of the reaction. The reaction includes the formation of a cyclic fructofuranosyl intermediate.

### 2.3. Synthesis of Levulinic Acid

Levulinic acid and its derivatives are now considered as an important intermediate in the conversion of carbohydrates. High-value chemicals of this type can be prepared selectively from cellulose under microwave irradiation (Table 1) [25]. Microwave-assisted depolymerization produced liquid products from cellulose with the yield of 87% using acid catalysts. The selectivities to glucose or levulinic acid (Figure 2) were 50% and 69%, respectively. Other researchers disclosed MW-depolymerization of cellulose leading in a 55% yield of glucose (20 min) [26].

A novel one-step process for the synthesis of levulinic acid from furfural produced from natural carbohydrates over hierarchical zeolites in a microwave reactor has been reported [27]. Brønsted acid sites play a role in the catalytic activity and their high concentration and properly chosen strength are beneficial for the high yield of levulinic acid reaching 42% at 90% furfural conversion.

A ZnBr_2_–HCl system containing both Brønsted and Lewis acid sites was used for levulinic acid production from carbohydrates using microwave irradiation [28]. Glucose, molasses, and sucrose were used as feedstock for levulinic acid production. The reaction rate of glucose conversion to levulinic acid was much faster when both HCl and ZnBr_2_ were used together. The maximum yield of levulinic acid from glucose was 53%.

Catalytic microwave-assisted conversion of waste polysaccharides into methyl levulinate was found to occur efficiently with Al_2_(SO_4_)_3_ as a catalyst [29]. The yield of methyl levulinate enhanced by the use of microwave heating was about 40%.

Direct conversion of carbohydrates to methyl levulinate in methanol catalyzed by acid–base bifunctional hybrid zirconia–zeolite catalysts was disclosed [30]. At 180 °C, the yields of methyl levulinate from glucose, mannose and galactose, sucrose, starch, and cellulose, respectively, were around 67–73%, 78%, 53%, and 27%.

Microwave-assisted conversion of cellulose without catalysts at 200–280 °C was also studied [31]. The maximum yield of bio-oil (45%) was observed with amorphous cellulose as a starting material at 260 °C. Water present in the feed somewhat enhanced the yield by 7–8%. The introduction of charcoal as a microwave-absorbing material enhanced the yield of gas products. Levoglucosan was formed with a high yield by the microwave-assisted conversion of cellulose at 260 °C, i.e., at a much lower temperature than the thermal process occurring at T > 400 °C (Figure 3).

Microcrystalline cellulose is transformed with the yield of 100% into glucose under microwave treatment with NaOH serving as a catalyst [32] (Figure 4).

Microwave-assisted alcoholysis of cellulose to methyl levulinate in methanol catalyzed by SnCl_4_/H_2_SO_4_ was studied [33]. The highest yield achieved was 61.5%. Starch, sucrose, and inulin were also tested in comparison with cellulose in the optimized system and 48–60% yields of methyl levulinate were reached.

### 2.4. Production of Other Products under Microwave Activation of Carbohydrates

Lactic acid (CH_3_CH(OH)COOH) production from cellulose-containing raw materials such as normal corn starch, high-amylose corn starch, and waxy corn starch dissolved in water using microwave treatment has been investigated (Table 1) [34]. The lactic acid yield (53–55%) from waxy corn starch under the microwave irradiation at a shorter reaction time (0–5 min) was much higher compared to thermal heating. 

Ionic liquids in combination with microwave radiation provide beneficial conditions for carbohydrate conversion [35,36]. The microwave-assisted synthesis of 5-HMF from cellulose results in the yield of 5-HMF of 62% in 2 min using ionic liquids as the reaction medium. The Zr(O)Cl_2_/CrCl_3_ catalyst in 1-butyl-3-methylimidazolium ionic liquid is active in the synthesis of 5-HMF from cellulose using the microwave-assisted conversion [37]. Saha et al. [38] also studied cellulose conversion into 5-HMF under microwave conditions and used Zr(O)Cl_2_ as a catalyst. The process occurred in aqueous and biphasic solvents and the screening of metal chloride catalysts for dehydration of several carbohydrate substrates demonstrated that Zr(O)Cl_2_ is the most efficient catalyst. The yields of 5-HMF were up to 63% and 42% from fructose and glucose, respectively, in water using MIBK as the organic phase in a biphasic solvent. The use of a biphasic solvent containing MIBK and an ionic liquid 1-butyl-3-methylimidazolium chloride afforded the yield of 5-HMF up to 84% and 66% from fructose and glucose. Fructofuranose proved to be an intermediate. A mechanism for isomerization of glucopyranose to fructofuranose was proposed. The catalyst was recycled five times without any activity loss. ScCl_3_ in 1-butyl-3-methylimidazolium chloride was also shown to act as an efficient catalyst of this reaction [39]. 5-HMF was obtained in a high yield of 73.4% in 2 min with the microwave power of 400 W. The use of microwave irradiation not only reduced reaction time from hours to minutes, but also improved the 5-HMF yield as compared to the thermal heating.

Levulinic acid can be produced from cellulose selectively with the yield approaching 55% when the microwave reaction is performed in SO_3_H-functionalized ionic liquids [40] (Figure 5).

Microwave-assisted conversion of glucose and cellulose directly into 5-HMF occurs in ionic liquids with CrCl_3_ or ZrCl_4_ as a catalyst [41,42]. The yields are as high as 60% and 90% in the case of cellulose and glucose, respectively.

Two task-specific ionic liquids 1-allyl-3-methylimidazolium chloride and 1-(4-sulfobutyl)-3-methylimidazolium chloride were chosen for the depolymerization of starch-based waste into reducing sugars [43]. Up to 43% and 98% of reducing sugars were obtained using the latter ionic liquid at low temperature under microwave and low-frequency ultrasound modes of activation. The same authors used a combined microwave/ultrasound treatment for the hydrolysis of starch-based waste into reducing sugars to demonstrate a synergy of the two methods of activation that afford enforced heat transfer (microwave) and the intensive mass transfer (ultrasound) [44].

A cellulosic raw material was partially depolymerized in cholinium ionic liquid under the microwave irradiation (110 °C, 20 min) to improve the enzymatic hydrolysis [45]. Hydrolysis of cellulose under microwave conditions was also shown to proceed in the presence of an ion-exchange resin [46]. Numerous examples of the application of microwave treatment in hydrolysis of cellulosic wastes for plant raw material processing were described [47].

Cellulose can be effectively converted into esters under microwave heating in a *N,N*-dimethylacetamide/lithium chloride mixture with 4-(N,N-dimethylamino)pyridine as a catalyst [48].

Another approach to esterification of natural carbohydrates with parallel depolymerization has been proposed by Durange et al. [49]. This approach is based on hydrolysis/functionalization of biomass carbohydrates. Carbohydrates of sugar cane bagasse and Jatropha curcas cake were transformed with complete conversion via the acetylation reaction under microwave radiation with acetic anhydride and sulfuric acid as a catalyst. Polyacetylated carbohydrates were the main products of the combined process.

Pyrolysis of macroalgae or microalgae is now considered a robust source of chemicals and energy carriers [50]. To enhance the production of biofuels from algal biomass, advanced or non-conventional pyrolysis methods have been used. The use of catalysts in pyrolysis of algal biomass can reduce the formation of nitrogenates and oxygenates in the biofuels. Hydropyrolysis of algal biomass can generate biofuels with the energy content as high as 48 MJ/kg with a high yield of bio-oil up to 50 wt.%, which is about 30% higher compared to conventional fuels. Microwave-assisted pyrolysis of algal biomass significantly shortens the processing time due to the advanced and energy-efficient mode of heating. However, the use of microwave pyrolysis also favors the formation of bio-syngas by enhancing the yield of synthesis gas up to 84 wt.% depending on the feedstock used. The microwave heating in the presence of either Brønsted acid (H_2_SO_4_) or Lewis acid (CrCl_3_ or AlCl_3_) was also used for the conversion of carbohydrate-rich potato peel and sporocarps of the fungus Cortinarius armillatus to lactic acid. The maximum lactic acid yield of 62% was found at 180 °C, 15 min [50].

A summary of the most illustrative examples of the use of microwave activation in the conversion of mono- and polysaccharides is given in Table 1 with indication of the experimental conditions where it could be possible to find the information about the reaction temperatures, reaction times, and input microwave power.

It should be noted that there are only a few reports on the effect of the nature of the catalyst, though this issue may be equally important as the optimization of the microwave regime.

Microwave irradiation has also been used for the preparation of hybrid catalysts that can be applied in the conversion of carbohydrates. For instance, an interesting approach has been proposed by Russo et al. [51]. Reduced graphene oxide or carbon black was taken as a core and titanium dioxide was grown onto the core particles (8–9 nm anatase nanoparticles dispersed on the carbon surface). Benzyl alcohol was used as a dispersion medium. The catalysts of this type were tested in the production of furfural from renewable carbohydrates, in particular, d-xylose. Aqueous-phase dehydration of xylose into furfural occurred at 170 °C with high furfural yields reaching 67–69%, while the conversions were as high as 95–97%. The catalysts turned out to be stable under hydrothermal conditions and were stable towards deactivation by heavy residues in comparison to the known solid acid catalysts. The nature of the core particles (reduced graphene oxide or carbon black) did not influence the performance of the catalysts that much as far as the size of titania was kept at the level of 8–9 nm. Recyclization of these catalysts was demonstrated with washing and drying between the consecutive tests.

**Table 1 molecules-27-01472-t001:** Summary of the effects of microwave activation on the conversion of cellulose and other carbohydrates into valuable products.

Source	Product	Catalyst	Microwave Conditions	Advantages over the Thermal Process	Ref.
Synthesis of 5-HMF					
D-fructose, D-glucose, sucrose, maltose, cellobiose	5-HMF	Mesoporous TiO_2_	Water or DMSO, 5 min, 120 °C 300 W	5-HMF yield up to 54%	[11]
Carbohydrates	5-HMF	Hierarchical macro/mesoporous titanium phosphate	A few minutes at 100–120 °C	Increased 5-HMF yield up to 60%	[12]
Fructose, glucose, sucrose, cellobiose, and cellulose		Mesoporous tin phosphate	150 °C, 20 min	The yields of 5-HMF of 77, 50, 51, 39, and 32%, respectively, were obtained from fructose, glucose, sucrose, cellobiose, and cellulose	[13]
Glucose	5-HMF	AlCl_3_	A few minutes at 165 °C	5-HMF yield 70%	[14]
Carbohydrates	5-HMF	Bifunctional crystalline microporous organic polymers containing sulfonic acid and secondary amine groups	110–120 °C, 5–30 min,	Shortened reaction time (a few minutes)	[15]
Fructose	5-HMF	Highly ordered 2D-hexagonal mesoporous SBA-15 organosilica	135 °C, 20 min	Yield of 74% from fructose	[16]
Derivatives of fructose, glucose and cellulose	5-HMF	Tungstophosphoric acid encapsulated dendritic fibrous mesoporous silica	110–120 °C, 30 min, 150 W	Yield of 96.5% 5-HMF from fructose (microwave) vs. 18–41.5% (thermal)	[17]
Carbohydrates	5-HMF and levulinic acid	Sulfuric acid and montmorillonite clay	190 °C 200 W	The reaction time was reduced. Larger amounts of levulinic acid were found under microwave radiation vs. thermal treatment	[18]
Fructose, glucose, sucrose, and cellulose	5-HMF	ZrMo mixed oxides modified by carboxylic acids (stearic acid, palmitic acid, myristic acid, and lauric acid)	100–120 °C, 5–30 min	Reduced time of the reaction and lower reaction temperatures	[19]
Fructose	5-HMF	Ionic liquids	155 °C, 3 min at 240 W, 1 min at 400 W	The yield of 5-HMF up to 98% (microwave) vs. 80% (thermal). Decrease of the reaction time from 5 min to 1 min (microwave vs. thermal).	[24]
Cellulose	5-HMF	Ionic liquids	2 min	The yield of 5-HMF was 62% in 2 min using ionic liquids as the reaction medium	[35,36]
Fructose, glucose, cellulose	5-HMF	Zr(O)Cl_2_/CrCl_3_	300 W, 120 °C, 5 min	The yield of 5-HMF up to 84% and 66% from fructose and glucose	[37,38]
Cellulose	5-HMF	ScCl_3_ in 1-butyl-3-methylimidazolium chloride	2 min, 400 W	Microwave irradiation not only reduced reaction time from hours to minutes, but also improved the 5-HMF yield up to over 73% compared to the thermal heating	[39]
Glucose and cellulose	5-HMF	CrCl_3_ or ZrCl_4_	400 W, 2–3.5 min	The yields are 60% and 93% for cellulose and glucose under microwave conditions compared to thermal conditions (about 12%)	[41,42]
Synthesis of fructose and furfural derivatives					
D-glucose	D-fructose	Hierarchical BEA zeotypes	75 W, 75 °C, 0–2 min	TOF values increased by about 30% in microwave conditions compared to thermal mode	[20]
Xylose, water-insoluble hemicelluloses and water-soluble fraction of corncob	Furfural	Tin-loaded montmorillonite	600 W, 100 °C, 3 min	The furfural yield (76.79%) and selectivity (82.45%)	[21]
Xylan, D-xylose and D-lyxose	Furfural and other products	Heterogeneous molybdate catalyst	200–300 W, 10 s–10 min	The reaction time under microwave irradiation was reduced 400 times in comparison with the thermal process (from 25 h to 10 s to 5 min)	[22]
Synthesis of glucose					
Polysaccharides	Glucose	Polyoxometalates as catalysts	1 kW, 10 min, 120–220 °C.	The energy consumption in hydrolysis can be reduced by up to 23% using microwave activation	[23]
Microcrystalline cellulose	Glucose	NaOH as a catalyst	800 W, 20 min	The yield of 100%, the ratio of degraded solid residue was about 7 times higher in microwave compared to thermal regime	[32]
Cellulose	Glucose	Without catalysts	100–150 °C, 20–60 min	55% yield of glucose	[26]
Cellulose	Glucose or levulinic acid	H_2_SO_4_, NaOH, p-toluenesulfonic acid	140 °C, 2 h, 950 W	The yields of glucose or levulinic acid were 50 and 69%	[25]
Synthesis of levulinic acid and derivatives					
Furfural	Levulinic acid	Hierarchical zeolites	160–200 °C, 1 h	Yield of levulinic acid 42% at 90% furfural conversion	[27]
Glucose, molasses and sucrose	Levulinic acid	ZnBr_2_-HCl	6 min	Yield of levulinic acid from glucose was 53%	[28]
Cellulose	Levulinic acid	SO_3_H-functionalized ionic liquids	800 W, 160 °C 30 min	The yield 40–50%	[40]
Polysaccharides	Methyl levulinate	Al_2_(SO_4_)_3_	160 °C, 45 min	The yield of methyl levulinate enhanced by the use of microwave heating was about 40%	[29]
Carbohydrates	Methyl levulinate	Acid–base bifunctional hybrid zirconia–zeolite catalysts	140–180 °C, 3 min	The yields of methyl levulinate from glucose, mannose and galactose, sucrose, starch and cellulose, respectively, were around 67–73%, 78%, 53% and 27%	[30]
Cellulose	Levoglucosan	Without catalysts	300 W, 200–280 °C, 3 min	Levoglucosan was formed with a high yield by the microwave-assisted conversion of cellulose at 260 °C, i.e., at a much lower temperature than the thermal process (T > 400 °C)	[31]
Starch, sucrose, inulin, cellulose	Methyl levulinate	SnCl_4_/H_2_SO_4_	800 W, 180 ℃, 50 min	Yield up to 61.5%	[33]
Synthesis of lactic acid and other products					
Normal corn starch, high-amylose corn starch, and waxy corn starch	Lactic acid	Without catalysts	5–15 min, 135–165 °C	Yields are 53–55% from waxy corn starch under the microwave irradiation at a shorter re-action time (0–5 min) was much higher compared to thermal heating	[34]
Potato peel and sporocarps of the fungus Cortinarius armillatus	Lactic acid	H_2_SO_4_, CrCl_3_ or AlCl_3_	180 °C, 15 min	Lactic acid yield of 62%	[50]
Cellulose	Cellulose esters	4-(N,N-Dimethylamino)pyridine	90–450 W, 1–3 min	Reduced reaction time (from 2–24 h to 1–3 min)	[48]
Cellulose	Bio-oil	Without catalysts	200–280 °C	Yield of bio-oil 45%	[31]

A novel microwave-assisted hydrothermal route was developed for the synthesis of Zn^2+^/γ-Al_2_O_3_ materials active in the conversion of carbohydrates into 5-hydroxymethylfurfural [52]. The microwave-synthesized Zn/γ–Al_2_O_3_ catalysts demonstrated outstanding performance in the catalytic fructose dehydration resulting in the formation of 88% HMF at 120 °C for 2 h. The presence of Brønsted acid sites at the catalyst surface was shown to be important for reaching the optimum catalytic activity.

Carbon materials were produced by using a microwave pyrolysis approach [53]. Activated carbon samples were obtained by chemical activation of corn stover with H_3_PO_4_. The optimized pyrolysis conditions were as follows: the phosphoric acid to biomass ratio of 0.8, microwave power of 600 W, and reaction time of 20 min. Conversion of Douglas fir sawdust into bio-phenols was chosen to test the activated carbon materials. The bio-oils produced from the sawdust were rich in phenolic compounds (up to 90%) containing predominantly phenol, cresols, and ethylphenols. The phenolic compounds can be produced either by decomposition of lignin with the following deoxygenation or by degradation of carbohydrates via the intermediate formation of furfurals. The presence of acidic surface functional groups at the carbon surface was shown to be a key factor in reaching the best performance. The content of acidic phosphoric-containing surface functional groups in the catalysts was demonstrated to correlate with the microwave irradiation power.

A great number of examples can be found in the literature on the use of microwave irradiation in the conversion of cellulose-containing wastes. Such wastes include both municipal and industrial wastes, including those produced via biotreatment of municipal and industrial wastes [54,55,56,57,58,59].

Dominguez et al. [60] compared the efficiency of thermal and microwave pyrolysis of coffee hull pellets. The microwave power of 130, 270, and 420 W allowed the authors to reach the process temperature of 500, 800, and 1000 °C within 5 min. The microwave treatment produced a larger amount of gas compared to the thermal process. The H_2_ and synthesis gas contents in the microwave-assisted conversion were significantly higher than in the conventional mode of heating. It is important to note that the additives of char to the reaction mixture resulted in the occurrence of more intense gasification process under the microwave conditions.

Wang et al. [61] compared gas and char production from pine sawdust under the microwave and thermal heating regimes. The addition of charcoal under the microwave mode turned out to be efficient to reach a higher reaction temperature and higher gas make.

Fernández and Menéndez [62] used sewage sludge or coffee hulls to compare the conventional and microwave treatments. Increased gas yields with a high H_2_ and CO concentrations were characteristic for the microwave pyrolysis. The formation of hot spots under the microwave activation was postulated and thus the intensification of the gasification was explained.

Zhao et al. [63] used a pilot 18-kW microwave setup for pyrolysis of wheat and corn straws at a temperature of 600 °C. The liquid, gas, and solid yields were close to 1:1:1 under optimal microwave conditions.

Comparison of the efficiency of microwave and conventional thermal processing of different carbohydrate-containing feedstock is presented in Table 2. It is seen that the microwave approach provides, in general, better yields of bio-oil from coffee hull pellets and higher yields of hydrogen and/or synthesis gas in the conversion of straw and sawdust wastes. It is noteworthy that the production of CO_2_ is reduced in the case of the microwave activation of the pyrolysis process. There are numerous studies reporting on the lower temperatures for the microwave-assisted pyrolysis compared to the thermally activated process [64,65,66,67].

**Table 2 molecules-27-01472-t002:** Comparison of the microwave and conventional thermal processing of different carbohydrate-containing feed stock.

Source	Product Yield	Microwave Conditions	Thermal Heating	Reference
Coffee hull pellets	Oil yield, wt. %	9.80–13.57	7.90–9.19	[60]
Pine sawdust	H_2_ yield, vol. % CO_2_ yield, vol. %	16–32 6–28	0–22 10–53	[61]
Wheat or corn straw	Synthesis gas, vol. %	Over 55% of the total gas volume	Less than 40% of the total gas volume	[63]

Additives of microwave-absorbing materials like solid carbons and silicon carbide are very efficient in affording a fast heating of biomass [68,69,70,71,72]. This allowed one to boost the conversion of biomass and to improve the bio-oil or bio-gas yield as well as the overall process energy efficiency [73,74].

Of most importance are the results obtained in those cases when a microwave-absorbing additive acts simultaneously as an active catalyst accelerating one of the reactions occurring during the pyrolysis of cellulose-based biomass or in the course of the conversion of a specific carbohydrate into an value-added product. For instance, the use of four types of metal oxides (NiO, CuO, CaO, MgO) with loading of 3, 5, and 10 wt. % to sugarcane bagasse under the microwave processing at 500 W for 30 min resulted in different effects [75]. Whereas the addition of CaO or MgO (basic additives) enhanced gas formation, the use of NiO or CuO (redox additives) led to the production of a larger fraction of liquid products. Huang et al. [76] carried out pyrolysis of corn stover under N_2_ and CO_2_ atmospheres in the presence of NiO, CuO, CaO, and MgO (3, 5, and 10 wt. %) at 500 W for 30 min. Better performance was observed in N_2_ atmosphere. The increase in the metal oxide loading under N_2_ conditions resulted in a decrease of the reaction temperature, due to a catalytic effect.

The examples of the combined processing of coal and biomass are widely discussed in the literature. First, the addition of coal to cellulose wastes like any other carbon additive acts as a microwave absorbing agent, thereby reducing the reaction temperature and the overall energy consumption. Second, due to the presence of oxygen-containing species, such as carboxylic groups, at the surface of coal, a catalytic effect can be observed. Third, coal is usually a porous material and thus better dispersion and more uniform heating of biomass can be achieved. Fourth, co-processing of coal and biomass can reduce the release of some ecopollutants originating from coal, such as sulfur and nitrogen oxides (NOx, SOx), PolyAromatic Hydrocarbons, Volatile Organic Carbons, and Total Organic Carbons [77]. Finally, the use of coal together with biomass may reduce coal-based CO_2_ emissions (the carbon footprint) [78].

The use of hybrid nanomaterials makes it possible to further improve the performance of the carbohydrate processing. A few recent examples with metal-organic frameworks and coordination polymers that are known as robust and unique hybrid materials, especially their composites with carbon-based materials that improve the microwave absorbance characteristics, demonstrate the benefits of using high porosity and controllable acid–base properties in the conversion of carbohydrates.

One example of this kind is related to the conversion of biomass into 5-HMF under microwave conditions [15]. Porous polymers containing sulfonic acid and secondary amine groups provide reactive catalytic centers for performing the bifunctional heterogeneously catalyzed reactions of carbohydrates.

The use of mesoporous materials is beneficial for the conversion of simple carbohydrates, because the presence of mesopores guarantees the high specific surface are of the catalyst as well as the accessibility of active sites to reacting molecules. Mesoporous silica modified with acidic encapsulated moieties, such as heteropolyacid, in particular, tungstophosphoric acid encapsulated in dendritic fibrous silica [17], demonstrates extremely high activity in the conversion of both glucose and cellulose into 5-HMF: the catalyst was very active in fructose and glucose dehydration with the yield of 92–95% and 58% of 5-HMF, respectively, and full conversion of fructose at 120 °C for 30 min under the microwave heating conditions in different solvents. The 5-HMF yield from cellulose reached 16.2%.

Novel ordered two-dimensional hybrid materials based on hexagonal bi-functionalized mesoporous organosilica containing both Brønsted acid and base groups have been prepared and tested in the conversion of fructose into 5-HMF with the high product yield of 86% by using microwave irradiation [79].

An utmost important issue that arises after the discussion of mostly fundamental research works on microwave processing of carbohydrate waste is as follows: can the microwave activation (irradiation) be used on a commercial scale, i.e., in the industrial practice? To answer this question, let us consider one publication where a new concept of a commercial reactor system has been proposed, in particular, the microwave-assisted dual fluidized bed gasifier was put forward for the first time [80]. Three catalysts containing Fe, Co, and Ni metal nanoparticles supported onto an Al_2_O_3_ support were tested in biomass gasification and their efficiency was compared for their effects on syngas production and tar removal in the course of the gasification process. The data have shown that microwave activation is an effective heating method for biomass valorization. Ni/Al_2_O_3_ was demonstrated to be the most effective catalyst for synthesis gas production and tar removal with the gas yield exceeding 80%. The optimal ratio of the catalyst to biomass was found to be 1:5–1:3. The addition of steam was shown to result in the improvement of the gas production and syngas composition, i.e., a higher H_2_ content. The authors concluded that extremely high temperature (>1200 °C) can be easily reached with the use of the microwave heating method combined with cheap microwave absorbents, such as coal and other carbon materials or silicon carbide. This technology makes it possible to produce much cleaner gas than in lower-temperature region and reduce the energy consumption, which becomes much lower than that of traditional fluidized bed gasifier. In addition, the authors consider microwave heating as a mature technology and claim that the development of a commercial microwave heating system for biomass gasification is of low cost. The main idea of the new gasifier concept is to divide the fluidized bed into two zones, i.e., the gasification zone and the heating zone. A circulation loop of the bed material is created between these two zones. The circulating bed material acts as a heat carrier from the heating zone to the gasification zone. A basic adsorbent, like CaO, can be incorporated into the gasifier and mixed with the bed material for in situ CO_2_ absorption. Thus, a carbon footprint of the whole process will be significantly reduced. By using this concept, it is possible to produce a H_2_-rich synthesis gas with an extremely low tar content that can be used for cogeneration systems, Fischer–Tropsch synthesis or to feed fuel cells. Ni/olivine catalyst characterized by better mechanical properties (low attrition) compared to Ni/alumina can be used in the new concept of the microwave-assisted dual fluidized bed gasifier.

A pilot-scale microwave heating setup was constructed for the production of bio-oil from sewage sludge [81], and the effects of microwave process parameters and chemical additives on the quality and yield of bio-oils were studied. The pyrolysis temperature range was 200–400 °C. The authors concluded that it is technologically feasible to produce bio-oil by microwave-induced pyrolysis of sewage sludge by optimizing pyrolysis conditions and be choosing appropriate additives.

Unfortunately, no examples are available in the literature related to the continuous process of carbohydrates transformation into valuable products. Most examples are based on the use of batch-wise conditions and solutions of carbohydrates in water. If a continuous mode of operation would be realized, then the industrial application of the microwave technology would be more realistic. The authors believe that arrangement of a continuous process is quite possible, since the contact time of the reaction resulting in an almost complete conversion of the starting carbohydrate, even in the case of cellulose, is just 1–5 min and a flow of an aqueous solution through the microwave-heated reactor with a loaded heterogeneous catalyst is quite achievable.

## 3. Perspectives and Future Outlook

Obviously, the perspectives of the microwave-assisted catalytic transformation of carbohydrates, including depolymerization and direct conversion into value-added products such as levulinic acid, glucose, levoglucosan, 5-hydroxymethylfurfural, and other chemicals are related to the following tasks:The proper choice of the catalyst that should be both active/selective in the target conversion and capable of fast and controllable heating under microwave conditions by one of several possible mechanisms (dipolar polarization, conduction mechanism, interphase polarization, i.e., Maxwell–Wagner mechanism).The proper choice of the media for conducting the microwave-assisted transformation of a carbohydrate, this medium should be capable of: (1) dissolving the initial carbohydrate material and final product and (2) absorbing microwave irradiation so that fast heating and enhanced reaction kinetics may be realized, here ionic liquids are solvents of choice without any doubts.The proper choice of the reaction conditions to ensure the highest selectivity at the desired conversion and minimum energy consumption. There are many examples in the literature showing that too high impact microwave powers or improper use of active but poorly microwave-absorbing catalysts may eliminate the benefits of the microwave processing.

As a results of milder conditions created under the microwave regimes of the reactions and better selectivity for the process of carbohydrate processing compared to the conventional thermal regimes, the recyclization becomes possible, which was demonstrated in a number of publications as a smaller difference in the activity (conversion or yield) between the first cycle and the last cycle (usually 4–5 cycles have been studied) as compared to the thermal mode of operation (for instance, [15,17,21,30,42]).

The outlook of the future research into the effects of microwave irradiation on the conversion of carbohydrates is expected to be focused on the following trends:The use of solid municipal wastes containing a large fraction of cellulose as a packaging material.The processing of a wide range of different natural carbohydrate wastes (lignocellulose, sugar bagasse, and agricultural wastes).Deep studies of the application of hybrid catalysts and hybrid nanomaterials for the microwave-assisted conversion of carbohydrate wastes, such hybrid materials are expected to provide a synergy of the properties of the components including unexpected responses to the microwave irradiation applied in the course of processing.Attempts to combine microwave activation with other engineering solutions such as the use of supercritical fluids (CO_2_, fluorocarbons, and water). An attempt to use such a combination has been made by Patil et al. who studied in-situ ethyl ester production from wet algal biomass under microwave-mediated supercritical ethanol conditions [82].

## 4. Conclusions

Thus, the new approaches to carbohydrates conversion into value-added products include the use of microwave activation in combination with homogeneous or heterogeneous catalysts. The benefits of the use of microwave regimes for the processing of carbohydrates, both monosaccharides and polysaccharides, including cellulose, are related to: (1) reduction of the duration of the processes, (2) reduction of energy consumption, and (3) enhancement of the catalyst activity and, in some cases, selectivity to the target products by minimizing the formation of side products. It is noteworthy that the microwave processing can be scaled up to produce quite large batches of organic compounds (levulinic acid, glucose, and esters). The future studies are envisioned to unravel the unusual non-equilibrium effects caused by the microwave activation.

## Data Availability

Not applicable.

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
