# Peer review of "Microwave-Assisted Conversion of Carbohydrates"

_molecules, 2022, doi:10.3390/molecules27051472_

Round 1

Reviewer 1 Report

The present paper is an overview of existing literature dealing with the processing of carbohydrates assisted by microwave (MW) technology. The paper presents publications stating the improvement of carbohydrate processing by microwave technologies, and the incorporation of microwave technology in different types of processes including synergies with both heterogeneous and homogeneous catalysts. Comments are made on what makes a catalyst-system interact well with the microwave technology.

Carbohydrate processing is key in the utilization and recycling of a lot of waste and is therefore an important topic of investigation in relation to sustainability. Furthermore, platform chemicals from sustainable sources is important, and carbohydrate processing is a way of producing these.

General comments:

  • Many places it is stated that microwave technology is an improvement to the traditional thermal processes. However, it would be nice to illustrate with numbers comparing the process with and without the microwave technology. Thereby the MW improvements would be emphasized. I find this is essential for the paper to be published.
  • For section: Catalytic processes for conversion of carbohydrates based on the use of microwave irradiation it would be nice to have some sort of grouping of either the different raw materials, products, catalysts or operating conditions in tables to gain a better overview of the literature.

P1 L23: ’added-value’ should be ’value-added’

P1 L24: ‘recently became’

P1 L24-26: ‘The environmental aspect … utilized and recycled.’ Very difficult-to-read sentence. Should be rephrased. Eg ‘The environmental aspect of the problem is also important since municipal solid waste contains a large amount of polysaccharides (mainly cellulose) which should be utilized and recycled.’

P1 L29: ‘in the view of’ could be replaced with ‘considering’.

P1 L31: ‘instrument’ could be replaced with ‘technology’

P1 L32-33: Avoid using ‘we’

P1 L46: ‘realized’ can be replaced with ‘obtained’

P1 L47-48: How do you distinguish between increased reaction rates and increased production speeds?

P1 L51: ‘The microwave treatment…’ This sentence is very (too ?) favorable describtion of the merits of microwave treatment. At least some references should be given here to circumstantiate the statement.

P2 L65: Avoid using ‘We’. ‘… here most..’ should be ‘.. here the most..’

P2 L73: ‘the catalysts’

P2 Scheme 1: 5-HMF structure is poor quality.

P3 L115: AlCl3

P3 L119-121: which advantages and synergies do you refer to? Maybe elaborate on this.

P3 L140-145: How much better was the performance of the MW process?

P4 L148-152: How much was the performance enhanced using MW?

P4 L157: ‘polyoxometallates’ -> ‘polyoxymetalates’

P4 L161: ‘.. up to 23% while using MW activation’ -> ‘.. up to 23% using MW activation’

P4 L167: This is the first time you refer to scheme 2, but you also mention levulinic acid at P3 L139.

P5 Scheme 4: ‘ceuulose’ -> ‘cellulose’

P5 L209-210: ‘Microwave-assisted alchoholysis of cellulose to methyl levulinate in methanol catalyzed by SnCl4/H2SO4’. This sentence only makes sense if it is a headline. Add ‘was investigated’ or similar. Furthermore, you can consider changing ‘The highest yield of 65% was achieved’ to ‘The highest yield achieved was 65%’ or even ‘Microwave-assisted alchoholysis of cellulose to methyl levulinate in methanol catalyzed by SnCl4/H2SO4 was investigated, and the highest yield achieved was 65%.’

P5 L215: ‘was described’ -> ‘has been investigated’

P5 L231: abbreviation MIBK is explained, but MIBK is also mentioned earlier on P3 L110

P6 L137: ‘of’ should be ‘for’

P6 L238: This is the first time microwave power is mentioned. It might be relevant to include for the rest as well?

P6 L239: How much did MW improve the yield?

P7 L300-301: ‘The nature.. influence that much the performance of the catalysts..’ should be ‘The nature.. influence the performance of the catalysts that much..’.

P8 L336: ‘a=out’?

P8 L350: consider changing ‘Noteworthy that-…’ to ‘It is noteworthy that..’

P8 L368: ‘fou’ should be ‘four’

P9 L379: ‘he’ should be ‘the’

P9 L386: Is VOC not usually an abbreviation for volatile organic compounds rather than volatile organic carbons?

P9 L388: ‘coalbased’ should be ‘coal-based’.

P9 L411: ‘use’ should be ‘used’

P10 L457: ‘.. that should both active..’ should be ‘..that should be both active..’

P10 L459: ‘.. several of possible..’ should be ‘.. several possible..’

P10 L466-467: ‘The proper choice of the reaction conditions to ensure the highest selectivity of the conversion at the minimum energy consumption, there are many examples..’ should be rephrased. Eg: ‘The proper choice of the reaction conditions to ensure the highest selectivity at the desired conversion and minimum energy consumption. There are many examples..’

P10 L475: What is meant by ‘wide processing’? Is it the processing of a wide range of material?

P11 L489: The Conclusions section is very short and not very useful to the reader. Proper Conclusions should be added. Also, it is not appropriate to introduce new concepts such as ‘in-situ activation’, not discussed earlier in the paper, in the Conclusions.

Author Response

Responses to comments of Reviewer 1

First, we are very grateful to the reviewer for the valuable comments and we revised the manuscript by taking into account all these comments and recommendation. Below we present the comment and the response:

General comments:

Many places it is stated that microwave technology is an improvement to the traditional thermal processes. However, it would be nice to illustrate with numbers comparing the process with and without the microwave technology. Thereby the MW improvements would be emphasized. I find this is essential for the paper to be published.

For section: Catalytic processes for conversion of carbohydrates based on the use of microwave irradiation it would be nice to have some sort of grouping of either the different raw materials, products, catalysts or operating conditions in tables to gain a better overview of the literature.

Response: Table 1 was added to the manuscript. It presents comparison of the data obtained for different sources of carbohydrates and different catalysts.

P1 L23: ’added-value’ should be ’value-added’

Response: Correction is made

P1 L24: ‘recently became’

Response: Correction is made

P1 L24-26: ‘The environmental aspect … utilized and recycled.’ Very difficult-to-read sentence. Should be rephrased. Eg ‘The environmental aspect of the problem is also important since municipal solid waste contains a large amount of polysaccharides (mainly cellulose) which should be utilized and recycled.’

Response: Correction is made

P1 L29: ‘in the view of’ could be replaced with ‘considering’.

Response: Correction is made

P1 L31: ‘instrument’ could be replaced with ‘technology’

Response: Correction is made

P1 L32-33: Avoid using ‘we’

Response: Correction is made

P1 L46: ‘realized’ can be replaced with ‘obtained’

Response: Correction is made

P1 L47-48: How do you distinguish between increased reaction rates and increased production speeds?

Response: The term “production speed’ was replace by “productivity (or space-time yield)” (p. 1).

P1 L51: ‘The microwave treatment…’ This sentence is very (too ?) favorable describtion of the merits of microwave treatment. At least some references should be given here to circumstantiate the statement.

Response: The references have been added to this sentence

P2 L65: Avoid using ‘We’. ‘… here most..’ should be ‘.. here the most..’

Response: Correction is made

P2 L73: ‘the catalysts’

Response: Correction is made

P2 Scheme 1: 5-HMF structure is poor quality.

Response: Correction is made

P3 L115: AlCl3

Response: Correction is made

P3 L119-121: which advantages and synergies do you refer to? Maybe elaborate on this.

Response: The following phrases were added on page 3: “The advantages over the non-hybrid materials may be related to a combination of meso- and macroporosity, as well as to a combination of the presence of several functions (like acid and base functions, metal oxide or metal nanoparticle functions) exhibiting different but in some case complementary catalytic properties. The synergy of hybrid nanomaterials, inter alia, may originate from the strengthening of the functional groups of one component of the hybrid material due to the presence of another component.”

P3 L140-145: How much better was the performance of the MW process?

Response: The following phrases were added on page 4: “The reaction time was reduced 3-5 times by using the microwave mode of operation.”

P4 L148-152: How much was the performance enhanced using MW?

Response: This example was added with the extension: “in particular, reduced time of the reaction and lower reaction temperatures,”

P4 L157: ‘polyoxometallates’ -> ‘polyoxymetalates’

Response: Correction is made

P4 L161: ‘.. up to 23% while using MW activation’ -> ‘.. up to 23% using MW activation’

Response: Correction is made

P4 L167: This is the first time you refer to scheme 2, but you also mention levulinic acid at P3 L139.

Response: Correction is made

P5 Scheme 4: ‘ceuulose’ -> ‘cellulose’

Response: Correction is made

P5 L209-210: ‘Microwave-assisted alchoholysis of cellulose to methyl levulinate in methanol catalyzed by SnCl4/H2SO4’. This sentence only makes sense if it is a headline. Add ‘was investigated’ or similar. Furthermore, you can consider changing ‘The highest yield of 65% was achieved’ to ‘The highest yield achieved was 65%’ or even ‘Microwave-assisted alchoholysis of cellulose to methyl levulinate in methanol catalyzed by SnCl4/H2SO4 was investigated, and the highest yield achieved was 65%.’

Response: This sentence was revised as follows: “Microwave-assisted alcoholysis of cellulose to methyl levulinate in methanol catalyzed by SnCl4/H2SO4 was studied [33]. The highest yield  achieved was  61.5%.”

P5 L215: ‘was described’ -> ‘has been investigated’

Response: Correction is made

P5 L231: abbreviation MIBK is explained, but MIBK is also mentioned earlier on P3 L110

Response: Correction is made

P6 L137: ‘of’ should be ‘for’

Response: Correction is made

P6 L238: This is the first time microwave power is mentioned. It might be relevant to include for the rest as well?

Response: We added this information where it was possible, because not all the publications contain such information, only a maximum power of the unit is usually indicated.

P6 L239: How much did MW improve the yield?

Response: The new Table 1 provides the overview of the most interesting results reporting the selective production of single products, rather than a rich mixture of all possible products, including gas, liquid and solid products. Also, where it is possible, the comparison with the thermal processes is given. Unfortunately, many publications contain only the results obtained under microwave conditions without paying attention to a comparison with thermal processes.

P7 L300-301: ‘The nature.. influence that much the performance of the catalysts..’ should be ‘The nature.. influence the performance of the catalysts that much..’.

Response: Correction is made

P8 L336: ‘a=out’?

Response: Correction is made

P8 L350: consider changing ‘Noteworthy that-…’ to ‘It is noteworthy that..’

Response: Correction is made

P8 L368: ‘fou’ should be ‘four’

Response: Correction is made

P9 L379: ‘he’ should be ‘the’

Response: Correction is made

P9 L386: Is VOC not usually an abbreviation for volatile organic compounds rather than volatile organic carbons?

Response: Correction is made

P9 L388: ‘coalbased’ should be ‘coal-based’.

Response: Correction is made

P9 L411: ‘use’ should be ‘used’

Response: Correction is made

P10 L457: ‘.. that should both active..’ should be ‘..that should be both active..’

Response: Correction is made

P10 L459: ‘.. several of possible..’ should be ‘.. several possible..’

Response: Correction is made

P10 L466-467: ‘The proper choice of the reaction conditions to ensure the highest selectivity of the conversion at the minimum energy consumption, there are many examples..’ should be rephrased. Eg: ‘The proper choice of the reaction conditions to ensure the highest selectivity at the desired conversion and minimum energy consumption. There are many examples..’

Response: This phrase was corrected according to the recommendation of the reviewer: “The proper choice of the reaction conditions to ensure the highest selectivity at the desired conversion and minimum energy consumption. There are many examples…”

P10 L475: What is meant by ‘wide processing’? Is it the processing of a wide range of material?

Response: This sentence was revised as follows: “The processing of a wide range of different natural carbohydrate wastes (lignocellulose, sugar bagasse, agricultural wastes).”

P11 L489: The Conclusions section is very short and not very useful to the reader. Proper Conclusions should be added. Also, it is not appropriate to introduce new concepts such as ‘in-situ activation’, not discussed earlier in the paper, in the Conclusions.

Response: The conclusions have been revised. The advantages of the microwave processing have been outlined.

Reviewer 2 Report

Section  2

Lines 42-43

 The effects resulted from microwave irradiation with the frequency ranging from 200 MHz to 300 GHz are of particular interest.

Definition of the microwave frequency range is too wide, misleading. The frequency of the microwave source (e.g., 2.45 GHz) needs to be specified. Sentence must be reformulated. Add references

Lines 44-45

 The benefit of microwave activation is the energy supply via radiation but not by heat transfer or convection

This sentence needs to be reformulated. The benefit of microwaves is the rapid & selective energy supplied to molecules/species that absorb microwaves. This energy is also transformed into heat.

Line 54

The abbreviation MW must be defined in the lines above. Once defined, it needs to be used throughout the text

Lines 54-55

A catalyst or a MW-absorbing material (SiC, carbon materials) introduced into the reaction volume can be heated

This sentence needs to be reformulated. Not all catalysts absorb microwaves and SiC, carbon materials are examples of microwave absorbers.

Lines 63-66 need references to support

Generally, section 2 is not well organised

Section 3 should also discuss the microwave equipment/frequency and the catalyst recycling

Author Response

Responses to comments of Reviewer 2

First, we are very grateful to the reviewer for the valuable comments and we revised the manuscript by taking into account all these comments and recommendation. Below we present the comment and the response:

Lines 42-43

The effects resulted from microwave irradiation with the frequency ranging from 200 MHz to 300 GHz are of particular interest.

Definition of the microwave frequency range is too wide, misleading. The frequency of the microwave source (e.g., 2.45 GHz) needs to be specified. Sentence must be reformulated. Add references

Response: This sentence was revised as follows: “The effects resulted from microwave irradiation with the frequency ranging from 200 MHz to 300 GHz are of particular interest, although generally the frequency of 2.45 GHz is used [1-7].” Actually, there are a few works performed at different frequencies, but the results are difficult to compare. In general, the efficiency of microwave energy absorption is proportional to the square of the frequency, therefore, the higher the frequency, the lower input power can be used to reach the same effect.

Lines 44-45

The benefit of microwave activation is the energy supply via radiation but not by heat transfer or convection

This sentence needs to be reformulated. The benefit of microwaves is the rapid & selective energy supplied to molecules/species that absorb microwaves. This energy is also transformed into heat.

Response: This sentence was revised as follows: “The benefit of microwaves is the rapid and selective energy supplied to molecules/species that absorb microwaves. This energy is also transformed into heat.”

Line 54

The abbreviation MW must be defined in the lines above. Once defined, it needs to be used throughout the text

Response: Correction is made

Lines 54-55

A catalyst or a MW-absorbing material (SiC, carbon materials) introduced into the reaction volume can be heated

This sentence needs to be reformulated. Not all catalysts absorb microwaves and SiC, carbon materials are examples of microwave absorbers.

Response: This sentence was revised as follows: “A catalyst capable of absorbing microwave energy or a microwave-absorbing material (SiC, carbon materials) introduced into the reaction volume can be heated.”

Lines 63-66 need references to support

Response: References were added.

Generally, section 2 is not well organised

Response: To improve Section 2, we prepared Table 1 presenting the main results of the studies focused on microwave conversion of mono- and polysaccharides into valuable products, nit a mixture of gas/liquid/solid products.

Section 3 should also discuss the microwave equipment/frequency and the catalyst recycling.

Response: This part was revised by adding the following sentence: “As a results of milder conditions created under the microwave regimes of the reactions and better selectivity for the process of carbohydrate processing compared to the conventional thermal regimes, the recyclization becomes possible, which was demonstrated in a number of publications as a smaller difference in the activity (conversion or yield) between the first cycle and the last cycle (usually 4-5 cycles have been studied) as compared to the thermal mode of operation [for instance, 15,17,21,30,42],

Reviewer 3 Report

The review on microwave assisted activation of carbohydrates has potential to be published, but there are many points which must be improved prior it will be accepted. The authors have found a lot of interesting information in the literature, but they should spend more time on the restructuring of the article by making the appropriate conclusions.

(1) "Numerous literature data show that, under microwave activation conditions, higher reaction rates are achievable than under thermal activation" You should add a reference.

(2) "The microwave activation can be used more efficiently if the active catalyst components are placed on a support that is not a microwave absorbing material." I understand that authors are specialized in catalysis, but it does not mean that authors should not mention in the review article case studies where microwave irradiation was successfully used without any catalyst. I also believe that synergy between two fuels can be counted to the use of catalyst because one of the feedstock released catalytic materials for the conversion of the other feedstock "Effect of microwave and thermal co-pyrolysis of low-rank coal and pine wood on product distributions and char structure". The authors should emphasize with all these case studies that there is a huge advantage to use commercial heterogeneous catalysts during microwave treatment over case studies where mixing of different feedstocks or even without addition of catalysts the yields and product composition are sufficient for the application field. Because I do research on blue economy feedstocks and work on the publication where we don't use any catalysts, I know that it is possible to obtain very high yield of cellulose nanocrystals of the highest quality. Please, use good arguments in the introduction to convince me. I also miss good references in your introduction. There is a very good book chapter of V. Budarin and J. Clark " Low temperature microwave pyrolysis and large scale microwave applications" that shows there are different problems with the microwave activation with respect to the type of furnace which scientists use. In other words, there are not many good manufacturers for the microwave reactor and it has a huge impact on the conversion, possibly more than the selection of a catalyst.

(3) In the chapter 2 for the catalytic processes I would decrease the text and prepare schematic representation of all processes and catalysts use. Possibly an overview table with all references could be an options. It is a lot of information where the reader can get lost.

(4)  Why the authors decide not to mention the hydrothermal processes in the microwave reactor? I believe it is one of the most future orientated processes "Processing of Citrus Nanostructured Cellulose: A Rigorous
Design-of-Experiment Study of the Hydrothermal Microwave-Assisted Selective Scissoring Process". You should mention it and discuss potential catalysts which could be added to yield cellulose and any other value-added by-products.

(5) I would also recommend to split this large section 2 into several subsection. In one section, you could describe difference solvents, in another catalysts, and in the third one operating conditions. It is too much wording on one section.

(6) Prior you talk about prospective and outlook, I would recommend to add a section on the upscale and industrial use. are there any processes which are already commercially available? In the book of Budarin and Clark, there is a chapter related to large scale microwave reactors. You can also add something similar to your article.

(7) A section on techno-economic analysis or life cycle analysis would also benefit. What do numbers say about potential to commercialize one of the chemical processes?

(8) Conclusion should be expended. I believe that you have seen in your review significantly more interesting facts and potential. What is the main problem with the mcirowave activation and how could we solve these problems within a circular economy? It could have to do with the renewable energy sources, recycling of material, material and energy balance, etc.

Author Response

Responses to comments of Reviewer 3

First, we are very grateful to the reviewer for the valuable comments and we revised the manuscript by taking into account all these comments and recommendation. Below we present the comment and the response:

The review on microwave assisted activation of carbohydrates has potential to be published, but there are many points which must be improved prior it will be accepted. The authors have found a lot of interesting information in the literature, but they should spend more time on the restructuring of the article by making the appropriate conclusions.

Response: The new Table 1 provides the overview of the most interesting results reporting the selective production of single products, rather than a rich mixture of all possible products, including gas, liquid and solid products. Also, where it is possible, the comparison with the thermal processes is given. Unfortunately, many publications contain only the results obtained under microwave conditions without paying attention to a comparison with thermal processes. The conclusions have been also revised.

(1) "Numerous literature data show that, under microwave activation conditions, higher reaction rates are achievable than under thermal activation" You should add a reference.

Response: References have been added. Also, new Table 1 illustrates the benefits of the microwave approach.

(2) "The microwave activation can be used more efficiently if the active catalyst components are placed on a support that is not a microwave absorbing material." I understand that authors are specialized in catalysis, but it does not mean that authors should not mention in the review article case studies where microwave irradiation was successfully used without any catalyst. I also believe that synergy between two fuels can be counted to the use of catalyst because one of the feedstock released catalytic materials for the conversion of the other feedstock

Response: We included a couple of examples when no catalysts have been used (see, for instance Table 1 and the text related to references 24, 26, 31, 34, 35, 36.

"Effect of microwave and thermal co-pyrolysis of low-rank coal and pine wood on product distributions and char structure". The authors should emphasize with all these case studies that there is a huge advantage to use commercial heterogeneous catalysts during microwave treatment over case studies where mixing of different feedstocks or even without addition of catalysts the yields and product composition are sufficient for the application field. Because I do research on blue economy feedstocks and work on the publication where we don't use any catalysts, I know that it is possible to obtain very high yield of cellulose nanocrystals of the highest quality. Please, use good arguments in the introduction to convince me. I also miss good references in your introduction. There is a very good book chapter of V. Budarin and J. Clark " Low temperature microwave pyrolysis and large scale microwave applications" that shows there are different problems with the microwave activation with respect to the type of furnace which scientists use. In other words, there are not many good manufacturers for the microwave reactor and it has a huge impact on the conversion, possibly more than the selection of a catalyst.

Response: We agree with the reviewer that, even though literature reports effects of the microwave activation on the conversion, selectivity and other parameters of the processes that can reduce energy consumption, for instance, by decreasing the reaction temperature, the mere microwave technology is still lacking for the really efficient hardware and equipment. The recommended reference is included in the list. The review is focused on the catalytic processing of carbohydrates, preferentially into specific (single) target products, not a mixture of gas/liquid/solid products, although a few examples from this field are also presented. The authors are not experts in microwave physics and not technical specialists in construction of equipment, therefore, we cannot review the technical aspects like constructions of reactors and reactor technology. Nevertheless, a few references related to the upscaling and potential of industrial applications of microwave processing of carbohydrates are given. We should say that still very little is known about the effect of the frequency of microwave irradiation on the efficiency of such processes (as well as any processes occurring under microwave activation), since 99% of studies are performed using the standard frequency of 2.45 GHz, which is not optimal at all. Actually, there are only a few works performed at different frequencies, but the results are difficult to compare. In general, the efficiency of microwave energy absorption is proportional to the square of the frequency, therefore, the higher the frequency, the lower input power can be used to reach the same effect. But it is too early to discuss such effects as far as the data are very scarce.  

(3) In the chapter 2 for the catalytic processes I would decrease the text and prepare schematic representation of all processes and catalysts use. Possibly an overview table with all references could be an options. It is a lot of information where the reader can get lost.

Response: Table 1 was added to the manuscript. It presents comparison of the data obtained for different sources of carbohydrates and different catalysts. The new Table 1 provides the overview of the most interesting results reporting the selective production of single products, rather than a rich mixture of all possible products, including gas, liquid and solid products. Also, where it is possible, the comparison with the thermal processes is given. Unfortunately, many publications contain only the results obtained under microwave conditions without paying attention to a comparison with thermal processes.

(4)  Why the authors decide not to mention the hydrothermal processes in the microwave reactor? I believe it is one of the most future orientated processes "Processing of Citrus Nanostructured Cellulose: A Rigorous

Design-of-Experiment Study of the Hydrothermal Microwave-Assisted Selective Scissoring Process". You should mention it and discuss potential catalysts which could be added to yield cellulose and any other value-added by-products.

Response: The review is focused on the catalytic processing of carbohydrates, preferentially into specific (single) target products, not a mixture of gas/liquid/solid products. Also, we tried to limit the scope with more or less well identified sources of carbohydrates, therefore, we excluded a good deal of publications related to the conversion of lignocellulosics, municipal wastes, and wastes from fruits and vegetables, first of all because these sources are not quite well characterized in terms of the composition and structure and a wide range of products (not a single product) is formed during their processing.

(5) I would also recommend to split this large section 2 into several subsection. In one section, you could describe difference solvents, in another catalysts, and in the third one operating conditions. It is too much wording on one section.

Response: We tried to structure the chapter by the products, because it is very difficult to do this using solvents or catalysts.

(6) Prior you talk about prospective and outlook, I would recommend to add a section on the upscale and industrial use. are there any processes which are already commercially available? In the book of Budarin and Clark, there is a chapter related to large scale microwave reactors. You can also add something similar to your article.

Response: The authors are not experts in microwave physics and not technical specialists in construction of equipment, therefore, we cannot review the technical aspects like constructions of reactors and reactor technology. Nevertheless, a few references related to the upscaling and potential of industrial applications of microwave processing of carbohydrates are given.

(7) A section on techno-economic analysis or life cycle analysis would also benefit. What do numbers say about potential to commercialize one of the chemical processes?

Response: See the answer to the previous comments. We are not technical or economic experts and it is too early to make economic analysis before we have a clear picture of the benefits and effects of frequency, input power etc.

(8) Conclusion should be expended. I believe that you have seen in your review significantly more interesting facts and potential. What is the main problem with the mcirowave activation and how could we solve these problems within a circular economy? It could have to do with the renewable energy sources, recycling of material, material and energy balance, etc.

Response: The conclusions have been revised. The advantages of the microwave processing have been outlined.

Round 2

Reviewer 1 Report

The authors have responded to the points raised and may be published now.

Author Response

We are very grateful to the reviewer for the valuable comments and we revised the manuscript by taking into account all these comments and recommendations. 

Reviewer 2 Report

N/A

Author Response

(The authors gave the same response as above.)

Reviewer 3 Report

Some of the recommendations were integrated, but the article still needs significant improvements. The structure of the article is not appropriate. It does look good to cover all main findings in the Section 2. Authors must find the way how to visualize their results using drawings instead of the long table that is not well commented in their work. There is no connection between different paragraphs in Section 2, it is a number of ideas which remain undeveloped. Several sections which could include operating conditions, different types of catalysts, different products should be prepared and not one long section. If authors cannot find anything to the scale up, you could at least draw chemically what happens when different catalysts are added and discuss the potential of each catalyst with respect to the upscale. For the good review, at least 100 references are needed and you have less. The introduction is extremely short if you are not familiar with the technology, the reader will not continue to read it. Articles should be written NOT for a specialist, it must be written in the way that everyone who downloads it will be able to read it. I recommend to extend introduction and explain why microwave reactor is a unique technique, but at the same time there is no large production of chemicals using that technology and provide the outlook of what you are going to cover in your review.

Author Response

Responses to comments of Reviewer 3

First, we are very grateful to the reviewer for the valuable comments and we revised the manuscript by taking into account all these comments and recommendation. Below we present the comment and the response:

Comments: Some of the recommendations were integrated, but the article still needs significant improvements. The structure of the article is not appropriate. It does look good to cover all main findings in the Section 2. Authors must find the way how to visualize their results using drawings instead of the long table that is not well commented in their work. There is no connection between different paragraphs in Section 2, it is a number of ideas which remain undeveloped. Several sections which could include operating conditions, different types of catalysts, different products should be prepared and not one long section. If authors cannot find anything to the scale up, you could at least draw chemically what happens when different catalysts are added and discuss the potential of each catalyst with respect to the upscale. For the good review, at least 100 references are needed and you have less. The introduction is extremely short if you are not familiar with the technology, the reader will not continue to read it. Articles should be written NOT for a specialist, it must be written in the way that everyone who downloads it will be able to read it. I recommend to extend introduction and explain why microwave reactor is a unique technique, but at the same time there is no large production of chemicals using that technology and provide the outlook of what you are going to cover in your review.

Response: First, we revised the introduction to make it easy for understanding by the broad audience of readers. Also, we introduced additional structure of the review by the type of a product. We added also discussion of results of Table 1 and structured this table. We tried to improve the logic of the presentation and discussion of the data on microwave processing of different types of carbohydrates in order to provide better connection of different paragraphs and develop more ideas on the use of microwave technology in the title applications. A special emphasis was made on the effect of the catalyst nature in relation to the microwave processing in order to explain the perspectives for upscaling of the technology.